Dear GAC organizers,

We are excited to hereby submit a proposal for a GAC, entitled **Generalization of information - integrative encoding or category-based inference?** In this proposal, we explore the issue of detailing the mechanisms behind generalization, and the need for a new theoretical and computational framework coupled with novel experiments.

How does the brain determine appropriate behaviour in novel situations? Most agree that we need to generalize information from related previous experiences, but the way in which we do this is highly debated. Two leading theories argue for two very different mechanisms behind such behavioural flexibility. *Integrative encoding* postulates that coactivating, integrating, and re-encoding a separate but overlapping memory with the representation of the novel experience can provide additional related information. *Category inference* instead takes the position that the brain constructs abstract categories based on regularities (e.g., appearance, experience, functionality), that can be used to infer a behavioural trajectory in novel situations. In fact, integrative encoding may simply provide the mechanism through which categorization arises. If so, how do we determine which memories to, and to not, integrate to form categories? Do we even need inference at all (and if so when?) or can we just generalize based on integrated representations? Do we perhaps use different strategies under different situations?

This proposal aims to bring together proponents of these opposing views to delve deeper under the surface of current axioms and experimental work. The GAC members have all committed to active discussion of theoretical/computational implications, and to the development and implementation of novel experimental work. One strength of this particular GAC is that members from different backgrounds have committed to novel experimental work: in humans using behaviour, EEG, and fMRI; and in monkeys using behaviour and neuronal recording. This should allow for a truly multidisciplinary and concerted effort, from researchers with different viewpoints, to bridge existing gaps and contradictions. At the end of the term we aim to provide a unified vision, based on a functional and computational understanding of the brain's beautiful ability to learn, generalize and deploy highly adaptive behaviours in novel situations.

The core team of this GAC proposal include the following collaborators:

Jessica Taylor (Postdoctoral researcher, ATR, Japan), Helen Barron (Postdoctoral fellow, University of Oxford, UK), Xiaochuan Pan (Assistant Professor, East China University of Science and Technology, China), Dasa Zeithamova (Associate Professor, University of Oregon, US), Masamichi Sakagami (Professor, Tamagawa University, Japan), & Aurelio Cortese (Principal Investigator, ATR, Japan)

We would also like to take this opportunity to express our strong appreciation for the Generative Adversarial Collaborations program. We too, believe that this is the way science will really move forward.

Thank you in advance for your time and consideration,

Jessica Taylor, Helen Barron, Xiaochuan Pan, Dasa Zeithamova, Masamichi Sakagami, Aurelio Cortese

Title: *Generalization of information - Integrative encoding or category-based inference?*

**Scientific question:** How do biological organisms generalize previously-learned information for adaptive behaviour in novel experiences? This broad question spans interdisciplinary fields - from decision-making and perception, psychology, memory neuroscience, to theoretical considerations in machine learning and artificial intelligence (Cortese et al. 2019). One prominent theory (integrative encoding) suggests that when new memories are being made, we activate information from overlapping memory representations so that information from both are integrated and reencoded together. Behaviour in novel situations can be guided by simple recall of information related to these extended associative links. Although this mechanism is very simple, computationally this process could become tedious given the inestimable extent of potential (and sometimes unnecessary) associative links that could be formed. A different theory (category-based inference) suggests a computationally simpler mechanism: that humans use abstract thoughts, such as functional categorization, to make their behaviour more efficient. Categories provide a logical structure through which information learned for one stimulus may be generalized to other stimuli (members of the same category). However, this theory requires much higher order, complicated, abstractions than does the former. One possibility is that both of these theories might be valid, with an organism implementing different strategies dependent on their current circumstances. However, to date there exists no study that has clearly dissected the contribution of both theories, nor made formal predictions on how they could or should differ in neural or behavioural terms.

**Background:** Support for the idea that overlapping memories can be coactivated, integrated, and reencoded (integrative encoding) was provided in a study by Shohamy & Wagner (2008). While overlapping pairs of associations were being learned, co-ordinated increases in hippocampal and midbrain BOLD signal were found; the strength of which related to subsequent generalization performance. Further support for this idea was found in the results of an experiment by Zeithamova et al. (2012) under similar experimental conditions. After a little experience, when one paired association was presented, the distributed neural representation of the overlapping association was found to be reactivated and predictive of subsequent generalization performance. This type of integrated encoding has also been proposed to occur on the fly (Shohamy & Daw, 2015), so that memories can be integrated and reencoded (not during earlier learning but) during the generalization process itself. Evidence for this was found in a study by Barron et al. (2013), where repetition suppression was used to show that related representations are simultaneously activated (in the hippocampus and mPFC) during online imagination of a novel concept.

While these findings fit well within an integrative encoding framework, it cannot be ruled out that they might instead (or additionally) reflect category and inference-based processes. For some of these studies, this is due to a potential confound that stems from multiple presentations of generalization stimuli. Since inference should be required only at the first occurrence of such stimuli (recall is sufficient thereafter), analysing these data together may have masked inference-related neural activity. Indeed, in a monkey study where a large number of generalization stimuli were presented, (rather than a smaller number shown multiple times) results were more indicative of category-based inference (Pan et al., 2008, 2014). Specifically, at the time of generalization, neurons in the monkey LPFC predicted

reward for novel stimuli based on their functional category membership. However, this does not mean that integrated encoding was not used here at all. This experiment involved neuronal recording from monkeys and therefore the whole brain could not be examined. Monkeys may have used integrative encoding in the hippocampus during learning or on the fly- possibly even to form categories- and then used this information for inference. Consistent with this idea, recent studies have suggested that categorization itself may arise from successful integrative encoding (Bowman & Zeithamova, 2018). However, the original proposal of integrative encoding argued that inference is not required because recall of integrated memories should suffice to guide behaviour. If this is true, then why do some studies show supposedly clear *inference* neural activity in higher order regions of the brain (e.g. Alfred et al. 2018; Pan et al., 2008, 2014; Wendelken & Bunge, 2010)? Is it possible that we sometimes generalize based on the simple recall of integrated representations but sometimes instead use inference? If so, how is this determined?

**Challenge or controversy:** When responding to novel stimuli, can we simply recall information from associative links that were created via mnemonic processes during learning? Or do we need to make inferences based on structured prior knowledge? Proponents of these two opposing theories show no consensus on the underlying mechanisms, even within each theory. For example, do association and generalization processes arise from mechanisms that occur during learning, recall, or both? The possibility that multiple strategies are valid and that which we use depends on individual circumstances/characteristics remains yet to be well explored. One complication concerns the fact that electrophysiology work in animals is not always directly comparable to human studies. Furthermore, besides obvious differences in previous experiments' design and the specifics of the tasks used, the lack of consensus may also stem from the absence of a convincing, comprehensive theoretical framework. Indeed, at present it is difficult to arbitrate between these views in that, although different theoretical and neuronal predictions exist, there are no obvious differences for predictions at the behavioural and computational level.

**Competing hypotheses and proposed approach for resolution:** *Competing Hypothesis 1:* Integrative encoding involves co-activation, integration, and re-encoding of separate memory traces in the hippocampus - if this occurs then (a) the process should be the same for each learned association, and (b) higher abstract thought is not required and so hippocampal, but not prefrontal activation should relate to generalization performance. *Competing Hypothesis 2:* Category-based inference involves the analysis of systematic regularities to define structured categories, addition of new items that fit within category constraints, and inference based on category membership - if this occurs then (a) the process should become more streamlined as distinct categories are formed, and (b) higher abstract thought- and therefore prefrontal activity- should relate to generalization performance. *Competing Hypothesis 3*: Category-based inference involves an integrative encoding process by which new objects/experiences are added to a categories and then subsequent inference is made based on category membership - if this occurs then (a) the process should be the same for each learned association, and (b) higher abstract thought- and therefore prefrontal activity- should relate to generalization performance. To resolve these questions, we plan to develop simple computational models reflecting each hypothesis, lay out individual predictions, and develop new (comprehensive) experimental designs to allow model validation and hypothesis testing at each stage of behaviour: learning, recall, and generalization (immediately, or after sleep, and using smaller and larger stimuli set for generalization). We will use a multiplexed

approach: combining behavioural testing and neuroimaging (fMRI and EEG) in humans, and behavioural testing and targeted neural recording in monkeys. This way, we plan to develop clearly testable hypotheses at (a) the behavioural level; e.g., Should reaction times differ depending on the theory and associated neural mechanism? Should task accuracy differ across conditions, such as when an appropriate behaviour can be correctly identified only by exclusion from an opposite one? And b) at the neural level; e.g., Is hippocampal or prefrontal activity (or a combination) more related to generalization performance? Is neural activity during learning or generalization (or a combination) more important for generalization? Are category or paired stimulus associations more easily decoded? And in which regions? Finally, we aim to collate the findings from these behavioural, neural and modelling/simulation lines of research into a unified vision.

**Concrete outcomes:** We expect this collaborative initiative to lead to the following outcomes, incrementally over the coming years. 1) lead to a concrete theoretical framework for the mechanisms underpinning generalization. 2) demonstrate a computational theory of how integrative encoding, category-based inference, and possible mixtures of the two may be implemented. 3) devise experiments that can clarify the most glaring contradictions, with support for either theory, or their mixture, in terms of behavioural and neural data. 4) open the way for future experiments and novel theories that may stem from a better understanding of the basic mechanisms of generalization - such as in the fields of memory, decision-making, metacognition, and learning. We also believe that, in the long-term, results gleaned from this collaboration will provide useful insight to address generalization in artificial agents, which remains a significant and unresolved hurdle.

**Benefit to the community:** We expect this GAC on the generalization of information through integrative encoding and/or category-based inference to generate a cascade of opportunities. First and foremost, the possibility to explore, at a significant philosophical and conceptual depth, a scientific issue that spans disciplines. Only through concerted effort by a group of scientists from diverse backgrounds will this conceptual and theoretical question be resolved. The implications of developing a novel theoretical framework of how generalization may be implemented by the brain - through memory integrative re-encoding or abstract category-based inference or their interplay - could have long-lasting ripples much beyond the respective traditional neuroscience fields. Possibly - and as is often the case - such knowledge is expected to leak into other domains to inform new experimental approaches and/or update existing formal theories. Finally, this GAC will create new bridges - beyond the single scientific topic of the proposal, among a network of investigators that spans 3 continents, with several at an early career stage. We expect this GAC to further support the training of early-career researchers, both through exposure to a novel and exciting way of doing research, as well as through extended discussion and thought-exchange (and dependent on the state of the current worldwide pandemic, visits) between the collaborating laboratories.

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

**Aurelio Cortese (Principal investigator, ATR - Japan)**: Co-organizer of the proposal and researcher. Aurelio will co-lead the collaborative effort, will be involved in identifying open questions, will devise computational models, contribute to task and experimental designs, and to collecting and analysing human behavioural and fMRI data.

Statements indicating commitment from each core group member to collaborate on the chosen GAC topic, including:

Each of the core members of this group, Jessica Taylor, Helen Barron, Xiaochuan Pan, Dasa Zeithamova, Masamichi Sakagami, & Aurelio Cortese, fully commit to (**a**) incorporating feedback from the community and potentially welcoming new CCN community members to the GAC based on their written commentary to the GAC proposal; (**b**) if our GAC is selected, participating in the running of an online kick-off workshop for CCN2020 in September or October, inclusive of both founding core GAC members and those new members who joined through the community feedback process; (**c**) if our GAC is selected, writing of the position paper that will clearly identify and lay out the concrete goals of the collaboration, to be submitted ~December 2020 to a curated special issue, and to be accompanied by commentary pieces authored by attendees of the CCN2020 kick-off workshop; (**d**) if our GAC is selected, attending and presenting all progress at the following CCN2021; (**e**) contributing to all of the above ethically and fairly, with particular emphasis on the growth of each individual and providing an inclusive space for all members' ideas and views, and with full transparency about any potential conflicts of interest.