# OpenReview forum: "Generalization of information - integrative encoding or category-based inference?"
_ccneuro.org/CCN/2020/Workshop/GAC_

### Official Review · ~Brian_Cheung1 · 2020-08-26
**A concrete framework/conditions for generalization based on previously acquired experience**

**Rating:** 8
**Soundness:** Agree
**Confidence:** 2

**Review:**

This proposal investigates the cause of generalization for biological neural systems. This is of significant interest to both the neuroscience and artificial intelligence communities. In particular, the utility of this type of learned generalization is only beginning to be discussed in the context of artificial agents. It is clearly written and if the experiments lead to clear distinctions between the integrative encoding theory and the category-based inference ideas, it would be of significant impact to theoretical models both in neuroscience and machine learning. Since much of the proposal revolves around generalization, a larger focus on how to objectively define generalization would make this proposal even stronger. Since it will be used as the main performance metric where conclusions will be drawn from, more expansion on how this will be concretely tested would be helpful.

**Comments:**

As the authors mention, the associative links proposed by the integrative theory do not seem in conflict with the prior structures required in category-based inference. This is where I think generalization will need to be more formally defined. Otherwise, one concern is that the results would not lead to a strong conclusion/distinction between these two theories occurring in the brain. Again, the authors also mention that a theoretical framework would be necessary to make integrative theory and category-based inference concrete enough to draw a well-defined conclusion. But I think it should go further and also have a strong theoretical framework (not necessarily a universally agreed upon one) for the definition of generalization and objective way to determine whether it has occurred or has not occurred. In lieu of a theoretical framework, an explicit list of the experimental conditions where generalization has or has not occurred would also be helpful.

**Controversy:**

Agree

**Definition:**

Neutral

**Expertise:**

Neutral

**Outcomes:**

Agree

---

> ### Public Comment · ~Aurelio_Cortese1 · 2020-09-08
> **Author reply**
>
> The suggestion for a clearer and explicit definition of generalization is of great value and well appreciated - in that, as the reviewer points out, it will help lay out how individual and global predictions made by each hypothesis may be fulfilled or not.
> While the next stages of this collaboration will provide the space and environment to discuss in depth this issue, and further elaborate on the specific experimental designs to be employed, we start by providing an initial working definition of generalization.
> Generalization is intended here as the transfer of knowledge to new, previously not experienced, situations/contexts/tasks. More specifically, the ability to perform better than chance, or higher than a certain baseline threshold, from the first trial(s) in a novel situation/context/task. Objective measures to determine whether it has occurred or not could therefore be: (i) accuracy - e.g. % correct, but also (ii) reaction times, (iii) degree of false memory formation (recognition memory), (iv) subjective confidence.
> In the context of this proposal and collaboration, two main lines of experimental approaches may be used.
> For behavioural testing, it may be helpful to use a combination of generalization tasks. These would combine categorization tasks with novel stimuli - where we can use categorization models (prototype vs exemplar) to differentiate people’s strategies; episodic inference tasks such as associative inference AB & BC -> AC, and transitive inference A > B, B > C, C > D, D > E, B ? D. It would also be of interest to see if participants can generalize appropriately in line with the requirements of the current task (e.g. dependent on whether they were told to focus on the color or shape of stimuli on the current trial) and if “appropriate generalization” is better supported by one or the other of these potential generalization strategies.
> For neural signatures, a powerful approach to distinguish between IE and CBI would be to test the representational differences of task stimuli. For this, it may be worthwhile to consider a setting with a single task / paradigm with lots of trials.
> The workshop will provide an excellent platform for the members of our group to further and more intensively discuss the specifics, with inputs from other members of the community. The goal, at the end of the workshop and for the position paper, will be to provide (i) an explicit list of definitions and the experimental conditions that we can use to test learning and generalization, (ii) behavioural measures and (iii) the neural signatures predicted under each generalization policy.

---

### Public Comment · ~Linda_Q_Yu1 · 2020-08-22
**Great topic, would love to learn more**

I'm very enthusiastic about the topic of this proposal, and think the problem discussed is an important one in the field of learning and memory. It does seem to me that whichever hypothesis would be true (integrative encoding, category-based inference, or one leading to the other) might depend on the stimuli and task requirements, e.g. whether they require a higher degree of abstraction. Are the authors saying that the predictions from their hypotheses would be true regardless of stimuli/generalization requirements, or would there be a graded effect? I would be interested to learn more about the parameters they are proposing to test these hypotheses within.

---

> ### Public Comment · ~Aurelio_Cortese1 · 2020-09-08
> **Author reply**
>
> Thank you for this important question and the enthusiasm for our proposal. We very much enjoyed the discussion this provoked within our group. In general, we feel that integrative encoding and category-based inference may be at the two ends of a spectrum of possible (interrelated) implementations, unlikely to be a purely either/or scenario. Task requirements/structure and stimuli do seem to influence the way in which generalization is implemented, and a strong candidate for the inconsistencies found in the literature. For example, Dasa Zeithamova from our group has a paper (in press) that provides evidence that some stimuli are suitable for creating an integrated memory while other stimuli are harder to integrate and seem to be encoded as separate traces1.
> Furthermore, it may be that different people use different strategies (unpublished data, Zeithamova lab) and that, in fact, different strategies may co-exist and both be represented within different regions of the brains of individual people2. The possibility that different strategies are encoded simultaneously in different neural circuits is something we are particularly keen to further investigate. In a paper to appear soon, Helen Barron from our group shows how the hippocampus and prefrontal cortex may be differentially involved in integrated and inferential codes. If this is true then what arbitrates between which strategy is used for behavioural responses and how does this arbitration differ between individuals? Are there cases where everybody uses the same strategy?
> Relatedly, the timing at which the integration of separate memory engrams may happen could be a further distinguishing factor - i.e., during learning / encoding (single day/session, multiple days/sessions, before/after sleep, etc.), or during generalization / retrieval. Will initially separate memories become integrated after inference is tested (for which there is some evidence already)3 ? Will the code used to support inference change over time and how might codes in different brain regions evolve after learning? For this purpose, we intend not only to investigate using a variety of stimuli and task parameters but also looking at cognitive performance at different points in time (during and after learning, before and after generalization, etc).
> In summary, these deliberations speak to the complexity of this topic in general and the importance (as pointed out by reviewers) of proper definitions of “generalization”, as well as of the different potential mechanisms underlying this. To the best of our knowledge, there is no existing empirical evidence which demonstrates that both integrative encoding (with a definition of simple memory links) and category-based inference co-exist in the brains of the same individuals.
>
> 1 Bowman, C.R. & Zeithamova, D. (in press). Training set coherence and set size effects on concept generalization and recognition. Journal of Experimental Psychology: Learning, Memory & Cognition.
> 2 Bowman, C.R. Iwashita, T. & Zeithamova, D (BioRxiv: https://www.biorxiv.org/content/10.1101/2020.05.26.117507v2)/ Model-based fMRI reveals co-existing specific and generalized concept representations.
> 3 Carpenter, A.C. &  & Schacter, D.L. (2019). False memories, false preferences: Flexible retrieval mechanism supporting successful inference bias novel decisions. Journal of Experimental Psychology: General, 147(7), 988-1004.

---

### Public Comment · ~Lennart_Bramlage1 · 2020-08-26
**Two theories that demand an integrated perspective**

Generalization (i.e., the use of prior knowledge for novel experiences) is a fundamental component of efficient learning and decision-making in biological and artificial organisms. The authors illustrate two distinct, popular theories on how this might be achieved: integrative encoding (IE) and category-based inference (CBI). While IE presents a compellingly simple mechanism of co-activation and re-encoding to integrate previously formed memories, it appears to be posing a hard search problem over possibly relevant associative links. Conversely, CBI generalization seems to be computationally less complex, but it requires well-defined abstract representations that don't seem to be well understood.

The authors make an excellent case for the possibility that both theories might not be at odds, but separate, even overlapping strategies in biological brains. The majority of the cited prior work forgoes such a holistic perspective, and previous attempts strike me as somewhat limited in their scope. It is indubitably clear that these considerations demand further research with a more comprehensive perspective.

Thus a few of my questions for the authors:
- IE is easy enough to understand at an implementation level, but the process of CBI is rather vague. Are the levels of analysis sufficiently aligned for a direct comparison?
- Following from the first question, is there a chance all elements of the CBI model may be implemented using purely associative learning rules? In other words: is it possible that IE is the implementing mechanism of CBI?

In conclusion, this is an excellent proposal that aims to bring together theories that compete at a surface level but may very well be interdependent and equally valid. The outlined course of action seems sensible to me. I would be especially interested in reading about the neural mechanisms of CBI and the behavioral implications of IE - effectively, the previously somewhat underreported side of each respective theory. I am very much looking forward to a unified framework of generalization in biological organisms.

---

> ### Public Comment · ~Aurelio_Cortese1 · 2020-09-08
> **Author reply**
>
> Thank you for the encouraging feedback.
>
> To question 1:
> The reviewer raised a critical point; essentially, how does abstraction occur in CBI? There are several (not necessarily mutually exclusive) processes that might underlie CBI. For perceptual categories (where membership likelihood is directly linked to perceptual properties) two processes that are often discussed are prototype versus exemplar-based categorization. Prototype-based categorization involves the averaging of different memories to create a “typical” representation of a category member. Newly encountered stimuli can simply be compared to this prototype. Exemplar-based categorization involves comparison of newly encountered stimul to existing members of the category (exemplars). The underlying processes may become more complex when we consider categories for which the members are functionally, but not perceptually, related (e.g. snacks, which vary a lot in the way they look and even taste, but can all function as something small to eat in between meals). However, when one considers neural evidence (e.g. Pan et al., 2008)1 we find that, after learning, certain neurons learn to fire in response to all members of functional categories, even when these are not perceptually related. In this case therefore it is likely the brain learns, via experience, to group stimuli in terms of their predicted outcome, rather than just on their perceptual features. Testing of these different processes can be implemented in monkey studies at the neuronal level, while in humans with fMRI using methods (repetition suppression or RSA/MVPA) that allow assessment of representational overlap between different cues.
>  At the computational level there are several ways we might implement this. We feel that reinforcement learning (RL) may be a useful framework here. RL could determine how the brain learns categories based on different kinds of similarity (perceptual and/or functional). To guide abstraction, the brain could use value and prediction errors. This does not necessarily mean that outcomes have to be explicit (reward or punishment can sometimes be intrinsic). In this way we can examine a simple mechanism that may underlie the formation of categories. RL might prove very useful in that it could also allow a value function to evolve over different representational levels. Consideration of other models in the literature, such as that by Koster et al. (2018)2, which suggests that recurrent loops within the hippocampus support pattern completion across multiple associations, might also be of use here.
>
> 1. Pan, X., Sawa, K., Tsuda, I., Tsukuda, M., & Sakagami, M. (2008). Reward prediction based on stimulus categorization in primate lateral prefrontal cortex. Nature Neuroscience, 11, 703 – 712
> 2. Koster, R. et al. (2018). Big-Loop Recurrence within the Hippocampal System Supports Integration of Information across Episodes. Neuron, 99, 1342–1354.
>
> To question 2:
> This is an important question and is a point of difference between members of this group and thus of central interest to probe further. On the one hand, integrative encoding (at least in terms of its definition in Shohamy et al., 2008)1 may not suffice to explain the implementation of CBI. For this to work, the original definition of IE might need to be extended so that not only single episodic memories, but also broader concepts (such as categories) are able to be re-activated and re-encoded with new experiences. Furthermore, even in this case, because it is unlikely that we associate every single thing that we experience together equally, some kind of structure and/or rules (e.g. based on visual/functional similarity and/or experience) are required to guide which concepts are and which are not combined via IE to form categories that allow for inference.
> On the other hand, however, integrative encoding has actually been proposed to be the implementing mechanism of a particular type of CBI2. According to this view, distinct episodic memories are co-activated and regularities assessed so that these can be combined to create and re-encode a prototype (against which new potential category members can be compared). Further experimental evidence is required to determine if IE may truly underlie (at least some types) of CBI, and if so, under which definitions of IE this works/does not work.
>
> 1. Shohamy, D., & Wagner, A.D. (2008). Integrating memories in the human brain: hippocampal-midbrain encoding of overlapping events. Neuron, 60(2):378-389
> 2. Bowman, C.R. & Zeithamova, D. (2018). Abstract memory representations in the
> ventromedial prefrontal cortex and hippocampus support concept generalization. The Journal of Neuroscience, 38 (10) 2605-2614.

---

### Public Comment · ~Hazel_K_Godfrey1 · 2020-08-26
**Pondering adaptive v maladaptive behaviour**

I would be very interested to see the authors' suggestion for concrete studies that could distinguish between evidence for integrative encoding or category based inference, or support for both depending on context.

As well as needing a interdisciplinary approach to answer the question "how do biological organisms generalize previously-learned information for adaptive behaviour in novel experiences", understanding the cognitive mechanism/s behind how behaviour is shaped by previous experiences would have broad implications for understanding and treating conditions where behaviour can be maladaptive, e.g., depression, anxiety, post traumatic stress disorder, chronic pain.

Under an integrative encoding account, behaviour may be maladaptive because the current information was not integrated with previously-learned information. Under a category based inference account, behaviour may be maladaptive because information is miscategoried. I'm sure the authors could think of other ways the system/s could break down too. Do the authors think consideration of how behaviour can be maladaptive would help to design studies to test what theory describes how biological organisms generalise previously-learned information, or to test if context drives which mechanism generalises previously learned information?

---

> ### Public Comment · ~Aurelio_Cortese1 · 2020-09-08
> **Author reply**
>
> Thank you for raising this important question, which we had not directly addressed yet. Consideration of this research theme through the lenses of psychiatric disorders is of great interest. Testing psychiatric populations may lead to valuable insights about the way IE or CBI functions are implemented and support adaptive behaviour. For example, in a different line of work, Vaghi et al.1 showed a dissociation between action and confidence in OCD patients. The confidence system was working as intended, but the output/computation was not used to drive behaviour. Similar issues may be found in psychiatric disorders in the case of IE / CBI.  Alternatively, we might find that these neural systems in psychiatric populations are working, but with some bias. For example, a hammer could be categorized as a tool or as a murder weapon, with the appropriate categorization depending on context. While the memory-integration and/or categorization systems of patients might be intact, it could be the case that context is not well used to determine the most appropriate interpretation for the given situation.
> Schizophrenia is an obvious condition to consider, where false inference occurs (paranoia, delusions etc). PTSD, where fear is generalized from fear-related to neutral stimuli/events might also prove an interesting avenue to explore (Please also see the reply to another commenter in relation to PTSD). As far as we are aware, high level cognitive tasks, such those that involve inferential reasoning, are rarely tested in these patients, and would surely prove very informative. Because this approach would require clinical collaboration, we need to consider this more and consider potentially extending this GAC to involve people from the relevant fields.
>
> 1. Vaghi, M.M. et al., (2017). Compulsivity Reveals a Novel Dissociation between Action and Confidence. Neuron, 96(2):348-354.

---

### Public Comment · ~Henrik_Daniel_Mettler1 · 2020-08-26
**How to generalize information of experiments on generalization of information?**

Gaining insight on how biological organisms adapt to experiences only loosely related to previous experiences seems like a key ingredient for a better understanding of the fast-learning and generalization capabilities that are thus far beyond the limits of what artificial agents can do.

Bringing together the seemingly opposing hypotheses of (1) relating new to previous experiences by overlap (integrative encoding) or (2) ordering new experiences into abstract thought categories, could potentially allow for valuable lessions on the neural underpinnings of how this generalization works.

I have a few questions to the authors regarding their approach:

- Generalization by integrative encoding seems to be closely related to memory recall in Hopfield networks on a single neuron level. How are the authors planning to relate their data with presumably lower spatial resolution (fMRI, EEG) to such models?

- The authors seem to hypothesize that hippocampal activity is more related to memory co-activation and prefrontal activity more to abstract thoughts/ categories. Is there empirical evidence to these assumptions?

- On a practical level, how are the authors planning to design their human-experiments such that they can ensure subjects do not rely on reactivation (resp. category inference) made upon memories (resp. categories) build prior to the experiment?

Overall I think this very exciting proposal brings together two interesting concepts on how generalization of information works and I am looking forward to see the outcomes of this research.

---

> ### Public Comment · ~Aurelio_Cortese1 · 2020-09-08
> **Author reply - part 1**
>
> Thank you for the encouraging feedback.
> To question 1:
> Although generalization/integration is supported by recruitment of overlapping sets of neurons at the cellular level, there are several prior studies that demonstrate that neural evidence which exists on the system level, as well as neural indices of integration, can be derived from fMRI1,2,3 and EEG4. Representational methods such as repetition suppression5, and MVPA6 ,7 have also proven useful in this line of research. We plan to run similar analyses with our own fMRI and hopefully EEG data. To supplement data in humans, we also have plans to run parallel single-cell recording studies in monkeys.
>
> 1. Shohamy, D., & Wagner, A.D. (2008). Integrating memories in the human brain: hippocampal-midbrain encoding of overlapping events. Neuron, 60(2):378-389
> 2. Bowman, C.R. & Zeithamova, D. (2018). Abstract memory representations in the
> ventromedial prefrontal cortex and hippocampus support concept generalization. The Journal of Neuroscience, 38 (10) 2605-2614.
> 3. Frank, L.E., Bowman, C.R., & Zeithamova, D. (2019). Differential Functional Connectivity along the Long Axis of the Hippocampus Aligns with Differential Role in Memory Specificity and Generalization. Journal of Cognitive Neuroscience. 1-18.
> 4. Varga, N.L. & Bauer, P.J. (2017). Using Event-related Potentials to Inform the Neurocognitive Processes Underlying Knowledge Extension through Memory Integration. Journal of Cognitive Neuroscience, 29(11), p.1932-1949
> 5. Barron, H.C., Dolan, R.J., & Behrens, T.E (2013). Online evaluation of novel choices by simultaneous representation of multiple memories. Nature Neuroscience, 16(10), 149201498.
> 6. Jiang, X., Chevillet, M.A., Rayschecker, J.P., & Riesenhuber, M. (2018). Training Humans to Categorize Monkey Calls: Auditory Feature- and Category-Selective Neural Tuning Changes. Neuron 98(2), 405-416
> 7. Linden, D., Oosterhof, N.N., Klein, C. & Downing, P.E. (2012). Mapping brain activation and information during category-specific visual working memory. Journal of Neurophysiology, 107, 628-639.
>
> To question 2:
> In line with our summary in the proposal itself, there is evidence for memory co-activation in the hippocampus1 and category-based inference in the PFC2. However, the exact contributions of each region to these different processes remains highly debated. Expanding on its traditionally proposed function of memory recall, the hippocampus may also contribute to acquired equivalence, associative inference and transitive inference (for a review, see Zeithamova, Schlichting, & Preston, 2012)3. Furthermore, the PFC has also been shown to represent co-activation of memories4. In fact, these two regions are often found to be coactivated in generalization tasks and many different reasons for this have been suggested; e.g. one proposal is that the hippocampus makes associations between stimuli that are then integrated by the PFC5. Clearly, a lot yet needs to be examined and clarified with respect to the roles of these two neural regions in generalization. Some of us suspect (as mentioned in response to another comment) that (rather than just one or the other) the brain might implement integrative encoding, category-based inference, and potentially other functions in between as well, meaning that these neural regions might have different roles dependent on which strategy is being used and under which context they are being tested. We hope to further probe these ideas within the scope of this GAC.
>
> 1. Shohamy, D., & Wagner, A.D. (2008). Integrating memories in the human brain: hippocampal-midbrain encoding of overlapping events. Neuron, 60(2):378-389.
> 2. Pan, X., Sawa, K., Tsuda, I., Tsukuda, M., & Sakagami, M. (2008). Reward prediction based on stimulus categorization in primate lateral prefrontal cortex. Nature Neuroscience, 11, 703 – 712
> 3. Zeithamova, D., Schlichting, M. L., & Preston, A. R. (2012). The hippocampus and inferential reasoning: building memories to navigate future decisions. Frontiers in Human Neuroscience, 6(March), 1–14.
>  4. Barron, H.C., Dolan, R.J., & Behrens, T.E (2013). Online evaluation of novel choices by simultaneous representation of multiple memories. Nature Neuroscience, 16(10), 149201498.
> 5. Wendelken, C., & Bunge, S.A. (2010). Transitive Inference: Distinct Contributions of Rostrolateral Prefrontal Cortex and the Hippocampus. Journal of Cognitive Neuroscience, 22(5), 837-847.

---

> > ### Public Comment · ~Aurelio_Cortese1 · 2020-09-08
> > **Author reply - part 2**
> >
> > To question 3:
> > At the moment we are considering the use of a battery of different generalization tasks, such as episodic inference, transitive inference, and acquired equivalence tasks. Similar to other tasks used in the literature, we can avoid the problem of participants’ pre-established memories/categories interfering with results by using completely novel stimuli and categories. For example, cartoon stimuli were used in one such experiment and these were separated into two groups based on features (such as type of tail) for which participants should not have had preconceived notions associated with these group labels1.
> >
> > 1. Bowman, C.R. & Zeithamova, D. (2018). Abstract memory representations in the
> > Ventromedial prefrontal cortex and hippocampus support concept generalization. The Journal of Neuroscience, 38 (10) 2605-2614.

---

### Public Comment · ~Toshinori_Chiba1 · 2020-08-26
**In relationship with PTSD**

I'm a medical doctor working on PTSD. PTSD is characterized by overgeneralization of fear- patients with this disorder generalize their trauma onto neutral stimuli.
Patients with PTSD have dysfunctional hippocampal function, a key region for the integrative encoding model, as well as dysfunctional PFC function, a key region for category inference. I wonder whether overgeneralization occurs due to either a dysfunctional hippocampus a dysfunctional PFC or both?
Deepening the understandings of this topic may directly aid in the development of new therapies for/treatment of overgeneralization of fear in PTSD.

---

> ### Public Comment · ~Aurelio_Cortese1 · 2020-09-08
> **Author reply**
>
> Thank you for your comment. Consideration of this research theme in terms of psychiatric disorders is of great interest. It seems plausible that overgeneralization” of fear could arise due to maladaptive “integration” of separate memories (e.g. co-activation of (1) a bomb’s loud noise with trauma, and (2) a car exhaust’s loud noise, so that these are re-encoded together to create an association (3) between the loud noise of a car’s exhaust and trauma). It also seems plausible that this could arise due to “mis-categorization” of neutral stimuli (e.g. the sound of a car’s exhaust becomes mis-categorized into the category of “loud and dangerous things”). It could very well be the case that this occurs in different ways for different patients (e.g. for patients with different subtypes of PTSD, or for patients with different types of trauma). Investigation into the mechanisms behind this “overgeneralization” could have broad benefits for the advancement of the neuroscience behind PTSD (particularly if the different neural mechanisms for each theory can be better defined so that maladaptive neural activity can be better understood), with potential implications for different types of therapy (e.g. do you need to re-activate the memories so that they can be re-encoded differently? Or do you need to train patients to categorize fear related/unrelated stimuli better?). This appears to be a very interesting line of related research, and we hope to pursue it further down the line through collaborations with clinicians.